# The Emerging Role of the Microbiota in Breast Cancer Progression

**DOI:** 10.3390/cells12151945

**Published:** 2023-07-27

**Authors:** Giancarla Bernardo, Valentino Le Noci, Martina Di Modica, Elena Montanari, Tiziana Triulzi, Serenella M. Pupa, Elda Tagliabue, Michele Sommariva, Lucia Sfondrini

**Affiliations:** 1Dipartimento di Scienze Biomediche per la Salute, Università degli Studi di Milano, 20133 Milan, Italy; giancarla.bernardo@unimi.it (G.B.); valentino.lenoci@unimi.it (V.L.N.); michele.sommariva@unimi.it (M.S.); 2Microenvironment and Biomarkers of Solid Tumors, Experimental Oncology Department, Fondazione IRCCS—Istituto Nazionale dei Tumori, 20133 Milan, Italy; martina.dimodica@istitutotumori.mi.it (M.D.M.); elena.montanari@istitutotumori.mi.it (E.M.); tiziana.triulzi@istitutotumori.mi.it (T.T.); serenella.pupa@istitutotumori.mi.it (S.M.P.); elda.tagliabue@istitutotumori.mi.it (E.T.)

**Keywords:** breast cancer, gut microbiota, mammary tissue microbiota, dysbiosis, progression

## Abstract

Emerging evidence suggests a profound association between the microbiota composition in the gastrointestinal tract and breast cancer progression. The gut microbiota plays a crucial role in modulating the immune response, releasing metabolites, and modulating estrogen levels, all of which have implications for breast cancer growth. However, recent research has unveiled a novel aspect of the relationship between the microbiota and breast cancer, focusing on microbes residing within the mammary tissue, which was once considered sterile. These localized microbial communities have been found to change in the presence of a tumor as compared to healthy mammary tissue, unraveling their potential contribution to tumor progression. Studies have identified specific bacterial species that are enriched within breast tumors and have highlighted the mechanisms by which even these microbes influence cancer progression through immune modulation, direct carcinogenic activity, and effects on cellular pathways involved in cell proliferation or apoptosis. This review aims to provide an overview of the current knowledge on the mechanisms of crosstalk between the gut/mammary microbiota and breast cancer. Understanding this intricate interplay holds promise for developing innovative therapeutic approaches.

## 1. Introduction

The human body, in particular the gastrointestinal tract, is populated by a large number of bacteria, viruses, fungi, and protozoa, constituting the so-called microbiota [1], also known as “forgotten organ” [2]. Over the past two decades, interactions between the gut microbiota and the host have been widely studied, highlighting its crucial role in a plethora of physiological and pathological processes. The most dominant phyla inhabiting the gut, about the 90% of the entire gut microbiota [3], are represented by Firmicutes and Bacteroidetes, but members of the Actinobacteria, Proteobacteria, Fusobacteria, and Verrucomicrobia phyla are also present [4]. These microorganisms establish a symbiotic relationship with the host and exert essential functions to preserve homeostasis. For instance, the microbiota is engaged in several metabolic pathways, such as the fermentation and absorption of undigested carbohydrates, and actively participates in energy harvesting, storage, and the activation and regulation of the immune system [5]. This delicate equilibrium can be subverted, and an imbalance between beneficial and potentially pathogenic bacteria has been observed in patients suffering from different pathologies. This condition, termed “dysbiosis”, results in drastic changes in microbial composition and is considered the effect of the microbial barriers’ disruption related to disease. However, recent studies have described how microbiota dysbiosis may represent the cause rather than the consequence of specific pathological conditions and/or events influencing the disease outcome [6]. For example, the gut microbiota has been demonstrated to be involved in promoting cancerogenesis, favoring tumor progression, and affecting the response to anticancer therapies, including immunotherapy [7].

The advent of high-throughput DNA sequencing technologies has made possible to characterize the entire human microbiome, that is, the collective genetic material of all the microorganisms living in our organism. The findings obtained using these novel techniques have allowed us to ascertain that, beyond the gut, other human body compartments, historically considered sterile, host indigenous bacterial communities [1]. Moreover, in a recent study conducted by Nejman et al., it was shown that tumors, including breast, lung, ovary, pancreas, melanoma, bone, and brain cancers, host their own microbiome, different from that present in the healthy counterpart [8]. It has been speculated that changes in bacterial abundance/composition in the tumor mass may represent an effect of the disease, related to the leakier vasculature in the tumor microenvironment that influences bacteria recruitment [9]. However, increasing evidence points to a “causal role” of tumor-associated bacteria in sustaining disease progression by shaping the phenotypes of cancer and immune cells and their interaction with the surrounding stroma. These data also raise important questions concerning the origin of these bacteria—whether they are tissue-resident or translocate from the gut or other sites in response to specific signals.

A specific microbiota is also associated with the mammary gland, once believed to be a microorganism-free environment [10]. The breast is mainly constituted by adipose tissue presenting an extensive vasculature and lymphatic drainage, and, for this reason, it represents a favorable environment for bacterial growth, particularly Proteobacteria and Firmicutes [11]. Culture experiments proved the existence of viable bacteria in the mammary tissue, revealing the colonization of *Bacillus* sp., *Enterobacteriaceae* sp., and *Staphylococcus* sp. [10]. Interestingly, a particularly rich and diverse microbiome has been identified in breast cancer [8]. Many studies have reported profound differences in the microbial composition of the mammary gland between tumoral and normal tissues and between benign and malignant tumors [8,12,13,14], supporting the notion that changes in tissues’ microbial communities may influence the progression of breast cancer [12].

Since both gut and local microbiota growing in the mammary gland have been postulated to influence breast cancer progression [14,15,16,17], in the present review, we aimed to give an overview of the state of the art regarding the intricate relationship between the gut- and mammary-tumor-associated microbes with the host in the onset and progression of breast cancer.

## 2. Relationship between Breast Cancer and the Gut Microbiome

Breast cancer is the second leading cause of cancer-related deaths in women worldwide [18]. It is a heterogeneous malignancy, and distinct molecular subtypes have been characterized. For example, the Luminal A subgroup is characterized by estrogen receptor (ER) expression and activity and, due to its good response to endocrine therapy, has the best clinical prognosis. Luminal B cancers express lower levels of ER and have higher proliferation rates compared to the previous subtype. The human epidermal growth factor receptor 2 (HER2)\positive subgroup is ER- and progesterone receptor (PR)-negative and comprises about 15% of all invasive breast cancers. It is more aggressive than luminal-like tumors. Finally, triple-negative breast cancers (TNBC), the subtype with the worst survival and the most challenging to treat, do not express hormone receptors or HER2 [19]. Such a classification represents an extremely valuable tool for predicting the clinical outcome and guiding the selection of the more appropriate therapy. However, it is becoming clear that other clinical variables need to be taken into consideration, and among these, gut microbiota is an emerging factor. Breast cancer’s occurrence and development has been demonstrated to be affected by the gut microbiota through different mechanisms, including the modulation of immune system activity, the alteration of estrogen levels, and the production of bacterial metabolites, which, in turn, exert various effect on tumor cells themselves and their microenvironment (Figure 1).

### 2.1. Gut Microbes–Immunity Crosstalk

The interaction between the commensal microbiota and the human immune system is in a dynamic balance [20]. Gut bacteria establish a complex and coordinated set of innate and adaptive immune responses to maintain tissues homeostasis. As a consequence, when the microbiota–host balance is disrupted and dysbiosis occurs, an increased production of inflammatory mediators, which are associated with cancer progression, is observed [21,22]. This effect has been experimentally demonstrated well in a mouse model of hormone receptor (HR)+ mammary cancer. Antibiotic-induced commensal dysbiosis resulted in a significant increase at the tumor site of myeloid cells highly expressing suppressive/inflammatory molecules, such as arginase-1 and IL-6 [23]. Among these inflammatory cells, M2-like macrophages, the most frequent immune subset in the breast tumor microenvironment and associated with reduced survival in HR+ breast cancer [24], were particularly found to infiltrate breast tumors and the normal-adjacent mammary gland during early and advanced stages of tumor progression. These effects were recapitulated by the fecal microbiota transplantation of dysbiotic cecal contents, demonstrating the direct impact of gut dysbiosis on mammary tumor growth [25,26].

Neutrophils have also been reported to be influenced by gut microbiota in the context of breast cancer [26]. In the C3-1-TAg mammary cancer mouse model, it has been observed that infection with *Helicobacter hepaticus*, a gut-resident bacterium, induced breast cancer progression associated with increased neutrophil recruitment and infiltration at the tumor site. Neutrophil depletion inhibited mammary tumor formation, resulting in the appearance of only a few pre-neoplastic and early neoplastic lesions in the breast tissue, as compared to multifocal advanced lesions in non-depleted mice [26].

Moreover, in a recent published study [27], it emerged that the specific gut bacteria is able to shape the immune response in a way that promotes or suppresses tumor development through the regulation of the stimulator of interferon gene (STING) agonists. In particular, the presence of cdAMP-producing *Akkermansia muciniphila* in the gut was observed to induce the IFN-I pathway upon STING activation. The IFN-I production led to the reprogramming of macrophages toward an anti-tumor phenotype and to the stimulation of the crosstalk between natural killer (NK) and dendritic cells (DC), further sustaining an anti-tumor immune response. Conversely, these events were halted in germ-free mice, in which it was possible to assist with monocytes’ differentiation into pro-tumoral macrophages [27].

The role of gut microbiota in breast cancer progression is also supported by clinical studies. It was found that low microbiome diversity was associated with the reduced survival of breast cancer patients. This condition was also accompanied by changes in immune cell compartments, consisting in a decreased level of lymphocytes and a parallel increased number of neutrophils [28]. Collectively, these findings clearly suggest that the gut microbiota may influence breast cancer progression and survival through the modulation of immune cells’ activity.

### 2.2. Modulation of Estrogen Levels 

Especially for HR+ breast cancer, the risk of breast cancer progression is highly associated with the level of circulating estrogens [29], whose metabolism takes place in the liver [30]. The role of estrogens as breast carcinogens has been proven by several epidemiological studies, and various mechanisms have been proposed to explain their pro-tumor effects. For example, upon binding to the specific nuclear receptor alpha (ER-α), estrogens are able to induce the enhanced production of growth factors that, in turn, boost the proliferation of breast cancer cells [31]. Moreover, estrogens have been reported to cause a genotoxic effect through a non-ER-α-dependent mechanism. Indeed, the catabolism of estrogens mediated by cytochrome P450 complexes generates reactive free radicals and intermediate metabolites that cause oxidative stress and genomic damage, resulting in increased mutation rates and a compromised DNA repair system [32,33].

C-18 steroid hormone estrogens exist in three biologically active forms, estradiol (E2, premenopausal), estrone (E1, postmenopausal), and estriol (E3, in pregnant women), which exert diverse biological effects. In the liver, the hydroxylation of parent estrogens E2 and E1 produces estrogen metabolites with varying hormone potency, bioavailability, and half-life. Estrogens and their metabolites are then conjugated through glucuronidation and sulfonation to allow biliary excretion into the gastrointestinal tract. A fraction of conjugated estrogens are deconjugated by the gut microbiota into free estrogens, which are then reabsorbed in the distal part of the intestine and, through the portal vein, distributed to other tissues, including the mammary glands [33]. Finally, they can circulate in the bloodstream as free molecules or bound to specific proteins. The link between estrogens and microbiota thus relies on the ability of intestinal bacteria to release free estrogens. In 2011, Plottel and Blaser widely discussed the “estrobolome”, defined as the collection of enteric bacterial genes with the ability to metabolize estrogens [34]. Indeed, free estrogens are mainly derived from the deconjugation process occurring in the gut via bacterial β-glucuronidase, especially microbial communities belonging to the Clostridia and Ruminococcaceae families or the Escherichia/Shigella genus. These β-glucuronidase-producing bacteria were frequently found to be over-expressed in dysbiotic microbiota due, for instance, to diet, alcohol consumption, and the use of antibiotics [35,36]. The augmented abundance of such microorganisms results in an increased concentration of circulating free estrogens, eventually contributing to breast cancer progression [37,38]. Accordingly, a case–control study conducted on 2266 North American women affected by breast cancer and 7953 healthy controls showed that women with a clinical history of long-term antibiotic treatment were characterized by an elevated risk of developing breast cancer [39]. In addition, adiposity, a condition strictly related to high circulating estrogen levels, has been associated with an elevated breast cancer risk in postmenopausal women [40]. A meta-analysis of 50 prospective observational studies also confirmed a relationship between adult weight gain and the risk of breast cancer in women [41].

### 2.3. Role of Microbial Metabolites 

Besides estrogens, other metabolic pathways link the gut microbiota to breast cancer progression. Through the metabolism and fiber fermentation of lipids or bile acids (BAs), bacteria produce an array of molecules that can directly or indirectly interfere with tumor cell proliferation [42,43].

Intestinal anaerobic bacteria, such as *Clostridia* spp., are one of the largest producers of lithocholic acid (LCA), a secondary bile acid found to decrease breast cancer cell proliferation through the activation of the G-protein-coupled bile acid receptor 1 (TGR5) [44,45]. Moreover, LCA is able to interfere with the mesenchymal-to-epithelial transition cell program, increase the tumor immune cell infiltration, and affect the tricarboxylic acid cycle (TCA) and the oxidative phosphorylation (OXPHOS) pathways [43]. Further in vitro studies demonstrated that LCA can decrease the expression of nuclear factor erythroid 2-related factor 2 (NRF2) and up-modulate Kelch-like ECH associating protein 1 (KEAP1), causing an imbalance between pro- and anti-oxidant enzymes, eventually impairing breast cancer cells proliferation [46]. Finally, LCA serum levels in breast cancer patients are associated with a high abundance of Clostridiales and Bacteroidales species. Early-stage breast cancer patients showed lower LCA levels and a concomitant reduced abundance of Clostridiales and Bacteroidales than healthy women [43].

Nisin, a gut bacteriocin produced by the Gram-positive *L. lactis*, has also been shown to have a highly cytotoxic effect on breast tumor cells by altering calcium ion influx across the cell membrane and promoting cell cycle arrest [47].

Other bacterial metabolites impacting breast cancer progression are short-chain fatty acids (SCFAs), such as butyrate, propionate, acetate, and lactate. SCFAs are the most common types of gut microbial metabolites, primarily produced by species colonizing the intestine, such as as *Eubacterium rectale*, *Clostridium leptum*, and *Faecalibacterium prausitzii*, as well as by the lactate-utilizing species *Eubacterium hallii* and *Anaerostipes* [47], through the fermentation of dietary fibers [48].

Evidence in breast cancer patients has shown that sodium butyrate has a promising anti-tumor activity on breast cancer cells alone or in combination with other anti-cancer agents [49,50,51,52,53], for example, the anti-HER2 antibody trastuzumab [54].

The diamine cadaverine, another bacterial metabolite derived from the decarboxylation of lysine and arginine, is known to inhibit breast cancer cells growth, migration and invasion, as well as suppressing epithelial-to-mesenchymal transition [45].

Overall, studies highlight how several bacterial metabolites can exert an anti-tumor effect against breast cancer cells. It should be considered that not all metabolites produced by the microbiota possess anti-cancer activity, but, instead, some of them are able to promote tumor growth, as shown in other cancer types [47,55]. However, to the best of our knowledge, no pro-tumorigenic bacterial metabolites have been identified in the context of breast cancer yet.

## 3. Breast Microbiome and Its Impact on Breast Cancer

Accumulating evidence indicates a consistent role of breast-tissue-resident bacteria in the onset and progression of breast cancer [14], but the origin of these bacteria remains unclear, and different hypotheses are debated. A study performed in canine breast tumors revealed the presence of bacteria belonging to the Bacteroides family in the tumor tissue, as well as in the mouth and gut [56], sustaining the notion of a possible bacterial translocation from the oral cavity to the intestine and, eventually, to the mammary tissue. However, the isolation from mammary tumors of bacteria typically inhabiting the skin, such as *Staphylococcus epidermidis* and *Micrococcus luteus*, shows a scenario in which these microbes may have reached the mammary gland through the nipples and then spread through the gland lobules and ducts [16].

Furthermore, live bacteria have been found within tumor cells and tumor-associated immune cells [8,15], suggesting that cancerous and host cells may be exploited as a shuttle to help microorganisms spread to the tumor or the adjacent normal mammary tissues [57]. Interestingly, Fu et al. also revealed that intracellular bacteria have the ability to induce the rearrangement of the breast cancer cell cytoskeleton, which confers tumor cells a higher resistance to fluid shear stress. This results in an increased survival rate during cancer cell transport through blood vessels and, consequently, an enhanced metastatic potential [15].

Several studies revealed modifications in the tumoral mammary gland microbial composition compared to the normal tissues and among tumors at different stages [8,12]. Urbaniak et al. reported that Enterobacteriaceae, Staphylococcus, and Bacillus were highly abundant in breast cancer patients compared to healthy individuals [14]. Xuan et al. [58] found the presence of *Sphingomonas yanoikuyae* in normal breast tissue and its dramatic reduction in the tumoral tissue, whereas the bacterium *Methylobacterium radiotolerans* was the most significantly enriched in the tumoral tissue. In an Asiatic cohort of breast cancer patients, Propionicimonas, Micrococcaceae, Caulobacteraceae, Rhodobacteraceae, Nocardioidaceae, and Methylobacteriaceae were enriched in tumors [59]. In the same study, a decrease was observed in the Bacteroidaceae family, and a parallel increase was observed in the genus Agrococcus as the malignancy developed. Moreover, cancer development also correlated with an augmented presence of Fusobacterium, Atopobium, Gluconacetobacter, Hydrogenophaga, and Lactobacillus genera [60].

In another study, Costantini et al. [61] reported that the most abundant genus found in the mammary tissue is represented by the bacterial genus Ralstonia, further increased in the breast tumoral tissue. Moreover, in the same study, the presence of Methylobacterium and Sphingomonas genera in the healthy mammary tissue was also observed, according to previous studies. Variations in the microbiota composition were also detected among the different breast cancer molecular subtypes. Banarjee et al. [62] firstly identified a unique microbial signature associated with triple-negative breast cancer. In a following work, the same authors defined four different microbial signatures associated with ER+, HER+, triple positive (ER+, PR+ and HER2+), and TNBC subtypes [13] (Table 1). The idea that each breast cancer molecular subgroup is characterized by a peculiar pattern of bacteria is also strengthened by Smith et al., who described a specific abundance of Eucaryarchaeota, Cyanobacteria, and Firmicutes phyla in TNBC [63]. These data support the notion that mammary dysbiosis, either being the cause or the consequence of tumor implantation, does occur in breast cancer and that changes in the microbiome are plausibly associated with its progression and with the intrinsic property of the specific subtype.

Moreover, in a study of 668 breast tumor tissues present in The Cancer Genome Atlas (TCGA) data set, the microbiome profile was correlated with the expression of specific tumor genes [64]. Interestingly, the presence of some bacteria, such as *Listeria fleischmannii*, was strongly associated with genes involved in the epithelial-to-mesenchymal transition, while *Haemophilus influenza* was correlated with pathways related to tumor growth, cell cycle progression, E2F signaling, and mitotic spindle assembly.

Collectively, these findings reveal that a peculiar tumor-associated microbiota composition can be associated with some features intrinsic to tumors. However, this type of study is still at its infancy and requires further investigation. For example, it is still unclear whether a correlation between specific bacteria and mutations harbored by breast cancer cells exists. This topic is particularly interesting considering that a genotoxic activity of *Escherichia coli*, Staphilococcus, and *Bacterioides fragilis*, isolated from breast tumors, has been clearly described [14,65].

**Table 1 cells-12-01945-t001:** Specific bacterial taxa found to be associated with the microbiota of triple-negative breast cancer tissue.

Taxa Enriched in TNBCs	Reference
Actinomycetaceae, Caulobacteriaceae, Sphingobacteriaceae, Enterobacteriaceae, prevotellaceae, Brucellaceae, Bacillaceae, Peptostreptococcaceae, Flavobacteriaceae	[62]
Actinomyces, Bartonella, Brevundimonas, Coxiella, Mobiluncus, Mycobacterium, Rickettsia, Sphingomonas	[66]
Azomonas, Alkanindiges, Caulobacter, Proteus, Brevibacillus, Kocuria, Parasediminibacterium	[67]

### 3.1. Mechanistic Role of Breast Microbiome in the Progression of Breast Cancer

As for the gut microbiota, recent studies have revealed various mechanisms through which local mammary-tumor-associated bacteria might play a role in breast cancer progression, including a direct carcinogenic activity, effects on cell growth or apoptosis, the modulation of the immune response, and the production of metabolites that, in many ways, can affect tumor biology (Figure 2 and Table 2).

#### 3.1.1. Carcinogenic Effect on the Host Genome

Urbaniak et al. compared the normal and cancerous breast tissues of patients undergoing mastectomy for breast reduction in the absence of neoplastic disease or for surgical resection of the tumor. In cancerous tissue compared to normal tissue, a higher abundance of Enterobacteriaceae and Staphilococcus was found [14], and subsequent culture experiments allowed the isolation of *Escherichia coli* and *Staphylococcus aureus*, two species belonging to the aforementioned genera. These bacteria are reported to possess a direct carcinogenic activity mediated by the production of colibactin, a genotoxin able to induce double-stranded DNA breaks and genomic instability [69,70]. Accordingly, the authors observed that HeLa cells exposed to *Escherichia coli* had significantly higher levels of histone-2AX phosphorylation, a marker of DNA double strand-break. A similar effect was also induced by Staphylococcus [14].

Moreover, a toxin from *Bacteroides fragilis*, a gut-colonizing bacterium also found in the mammary gland, can induce epithelial hyperplasia to promote tumor growth and metastatization via the β-catenin–Notch1 axis [65].

#### 3.1.2. Effect on Cell Growth/Apoptosis

One of the mechanisms regulating the crosstalk between microbes and the host is based on the expression of pattern-recognition receptors (PRRs), such as Toll-like receptors (TLRs), by different types of immune and non-immune cells. These receptors can sense microbial changes occurring in the tumor microenvironment and modulate the immune system activity and, in certain circumstances, tumor cell growth/proliferation [71]. For instance, it has been reported that *Fusobacterium nucleatum*, previously demonstrated to be associated with colorectal cancer (CRC) [72], is implicated in breast cancer growth [12,68] through the activation of TLR4/NF-kB pathway in cancer cells [73].

Moreover, it has been also demonstrated that this bacterium, through the Fap2 lectin protein, can bind Gal-GalNac, a sugar present in high levels, on breast tumor cells’ surfaces, causing an acceleration of breast cancer growth and the development of metastases [70,72].

#### 3.1.3. Effects on the Immunity

In a TNBC mouse model, we have recently observed the abundant presence of *Staphylococcus epidermidis* in the tumor niche [16]. In particular, *Staphylococcus epidermidis* was found to be responsible for an extremely inflamed tumor microenvironment, determined by its strong ability to induce pro-inflammatory cytokine secretion and complement activation, reported to sustain tumor growth [74]. The in vivo peritumoral transfer of this bacterium was also demonstrated to be associated with a significant increase in immunosuppressive T regulatory cells into the tumor nodules and, when co-cultured in vitro with bone-marrow-derived macrophages (BMDM), to promote a pro-tumor phenotype. Accordingly, antibiotic treatment, by the abundance of lowering *Staphylococcus epidermidis*, reduced tumor growth. The anti-tumor effect mediated by antibiotic treatment was accompanied by the appearance of *Micrococcus luteus* in the tumor mass. Unlike *Staphylococcus epidermidis*, *Micrococcus luteus*, when in vivo peritumorally transferred, exerted an anti-tumor activity by inducing an M1 macrophage phenotype and by reducing myeloid-derived suppressor cell (MDSC) infiltration [16]. These findings are in line with previously published data revealing the abundant presence of the Micrococcaceae family in healthy breast samples and of Staphylococcaceae in tumoral tissues [14].

Moreover, the bacterium Sphingomonas, detected in the healthy mammary gland, is able to induce the activation of invariant NKT (iNKT) cells [75], important mediators in cancer immunosurveillance and in the control of breast cancer metastases [76]. Accordingly, an increased level of Sphingomonas in healthy compared to tumoral mammary tissue has been observed to be associated with a higher expression of TLR2, -5, and -9 and of antimicrobial response effectors IL-12A, bactericidal/permeability-increasing protein (BPI), and myeloperoxidase (MPO), suggesting its possible protective role in cancer by sustaining immunosurveillance [58].

#### 3.1.4. Microbial Metabolites Production

It is still unclear whether tumor-infiltrating bacteria can produce metabolites, as largely demonstrated for gut microbiota. However, based on data present in the literature, it is possible to obtain some insights. For instance, *Bacillus cereus*, capable of metabolizing progesterone into 5-alpha-pregnane-3,20-dione (5αP) [77], was found to be higher in breast cancer patients than in healthy ones [10,14]. Since 5αP is believed to promote tumor development by stimulating cell proliferation, it is plausible to speculate that at least part of this molecule may be of bacterial origin [78].

Moreover, in a recent study performed in a cohort of patients with TNBC, a high abundance of Clostridiales in tumoral tissue was associated with an activated immune microenvironment [79]. Specifically, the presence of these bacteria positively correlated with the production of the metabolite trimethylamine N-oxide (TMAO), a compound able to activate CD8+ T cells-mediated antitumor immunity and M1 macrophages, further supporting the idea of a metabolically active tissue-resident microbiota [76].

## 4. The Gut–Breast Microbiota Axis

The existence of axes between gut microbiota and different body areas, such as the liver, lung, and brain, has already been reported, but no definitive proof is available today on the crosstalk between the gut microbiota and the mammary glands. However, it was observed that treatment with orally administered probiotics is highly effective in the cure of mastitis and that probiotics become detectable in human milk [80], strongly suggesting that an interconnection between gut microbiota and the breast may exist.

Gut-resident bacteria may leave the intestine through breaches in the epithelium, which is frequent during dysbiosis, and translocate to the mammary gland via the blood or lymphatic systemic circulation. An alternative escape route is represented by intestinal dendritic cells, which are reported to uptake bacteria in the intestinal mucosa through their ability to open the tight junctions between epithelial cells [81]. Since dendritic cells are migratory cells, they can reach distant sites, such as the mammary tissue, through the vascular system.

Furthermore, a third possible participant in the gut-microbiota–breast tissue dialog may be represented by bacteria metabolites. These bacterial products, produced in the intestine, may be absorbed by the intestinal mucosa and released into the bloodstream through which they can virtually reach all the body compartments, including mammary glands, exerting their biological functions in loco.

Although there are many insights regarding the possibility of a gut–breast axis, further investigations are still required not only to finally prove its existence but also to identify the players involved in their crosstalk.

## 5. Conclusions

A large symbiotic microbiota resides in the human intestine and exerts fundamental roles in health and disease. Since it is able to regulate host metabolism and shape the immune system, the gut microbiota has been revealed to affect breast cancer progression. More recently, bacteria have also been found to be a component of the breast mammary tissue, but their pathobiological role is poorly understood due to their low biomass. Many studies have clearly demonstrated that mammary tissue microbiota changes in the presence of a tumor, representing a scenario in which the tumor-associated microbiota actively participates in the constitution of the complex tumor microenvironment.

The present review summarizes what is known about the relationship between specific bacteria and breast cancer progression and, concomitantly, highlights what is still missing in the literature. Indeed, there are many open questions that represent weaknesses in this field: (i) Does a direct link really exist between the gut and the mammary microbiota composition? (ii) Are they seeded in the tumor microenvironment early on in tumorigenesis, or are they recruited as the tumor alters the microenvironment? (iii) Which bacteria can be defined as “good” or “bad”? (iv) What are the metabolic products or the structural molecules mechanistically involved in their effects on cancer cells? (v) Do different species share a common mechanism that is able to impact tumor cell biology and could represent potential therapeutic targets? A limitation of studies aiming to answer these questions is that the use of mouse models might not be exhaustive, as they do not allow one to consider many factors affecting the human microbiome, such as diet, host genetics, and age. Thus, only studies on human subjects may represent the future direction of microbiome research to deconvolute the complex microbiota–tumor crosstalk and to open new avenues to shape the cancer microenvironment toward a favorable context through the modulation of gut and/or local microbiota. Antibiotics, probiotics, prebiotics, and fecal microbiota transfer are strategies that are used to modulate the gut microbiome in the treatment of many infectious diseases and are currently investigated as potential anti-cancer therapeutic options.

## Figures and Tables

**Figure 1 cells-12-01945-f001:**
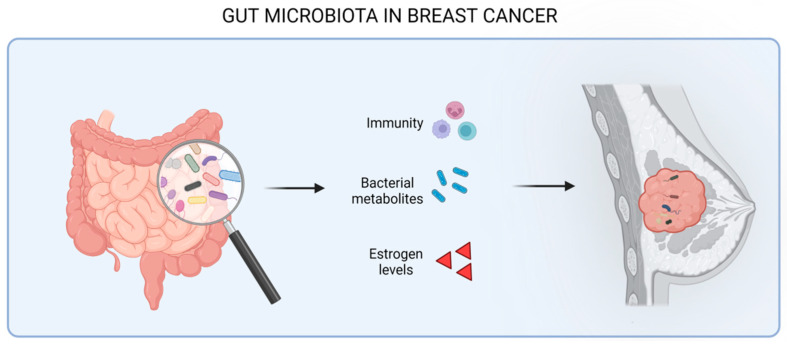
Effect of gut microbiota in the progression of breast cancer through the modulation of the immune system, the release of bacterial metabolites, and the modulation of estrogen levels.

**Figure 2 cells-12-01945-f002:**
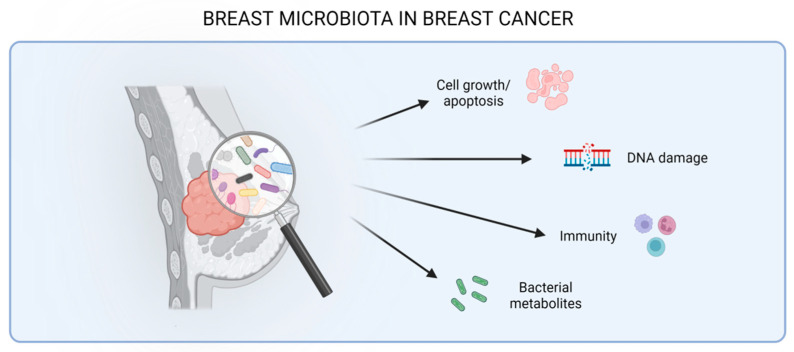
Effect of the mammary microbiota in breast cancer interfering with cellular pathways (growth or apoptosis), inducing DNA damage, modulating the immune system, and releasing bacterial metabolites.

**Table 2 cells-12-01945-t002:** Bacterial species found to be associated with the microbiota of breast cancer tissue, in patients, and in murine models and their function in the progression of breast cancer.

Breast Cancer Tissue	Bacteria	Molecular Mechanism	Reference
Human	*Escherichia coli* and Staphylococcus	Induction of DNA double-strand break and genomic instability in vitro	[14]
Human	Clostridiales	Inhibition of tumor growth by producing the metabolite trimethylamine N-oxide (TMAO) that activates CD8+ T cells- mediated antitumor-immunity	[53]
Human	*Fusobacterium nucleatum*	Breast tumor progression and metastases by fap-2 dependent binding of the bacterium to breast cancer tissue Gal-GalNac	[68]
Mice	Staphylococcus, Lactobacillus and Streptococcus	Breast tumor lung metastases by modulating the stress response and influencing cancer cell viability, altering the cell cytoskeleton	[15]
Mice	*Staphylococcus epidermidis*	Increased T regulatory cell infiltration in the tumor and complement pathway activation in vivo, and increased pro-tumoral M2 macrophages phenotype in vitro	[16]
Mice	*Micrococcus luteus*	Reduction of mammary tumor growth in vivo, and increased anti-tumoral M1 macrophage phenotype in vitro	[16]
Mice	*Bacteroides fragilis*	Breast tumor progression and metastasis through the secretion of the *B. fragilis* toxin (BFT)	[65]

## Data Availability

Not applicable.

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
