# Peer review of "The Emerging Role of the Microbiota in Breast Cancer Progression"

_cells, 2023, doi:10.3390/cells12151945_

Round 1

Reviewer 1 Report

The authors have done a good and very interesting literature review. However, The review does not adressed the question of what relationship there is between the gut and mammary microbiota. Also, English should be revised and there are several flaws.

line 59 --> This statement needs to be supported by a bibliographic citation.

line 83 --> It should use a more current reference (reference 17). 

Line 91-92 --> TNBC is not considered a rare tumor as it occurs in approximately 15% of all breast tumors. 

Figure 1B should appear with Figure 1A. I came up to expand section 2.2 to introduce the figure. 

Line 112 --> Change "that associate" to "that are associate".

Line 133 --> The sentence is not understood

Line 142--> The sentence should be changed. If the breast tumor already exists, it is not the risk of developing it that we are talking about.

Line 149--> It needs a reference. 

Line 159-162--> what is the role of estrogens in the gut? which molecules are considered free estrogens?

The authors should specify more how estrobolome can induce breast cancer progression.

Line 189--> Change "antitumor" to "tumor".

Line 209 and 216--> Instead of BC put breast cancer since they have not named those acronyms before.

In section 2.3 it only talks about metabolites that produce an anti-tumor effect. Is there no metabolite that produces tumor progression?

line 222--> Reference 53 does not speak of that. 

Line 226-227--> Modify the sentence. It is not understood

Line 256--> The sentence is not understood.

It would be interesting to introduce if the presence of bacteria are able to modify the genome of tumor cells.

Line 263--> change TN breast cancer to TNBC

Table 1 should be introduced in section 3.

The title of section 3.1.1 is not well expressed. 

Line 282--> "absence of a disease....." delete the "a".

Line 284 --> change to neoplastic tissu

Line 301--> The phrase is not understood.

Line 316--> change to "in a TNBC mouse model".

Line 323--> Not understand the phrase line 323--> No understanding of the phrase line 323--> Change to "in a TNBC mouse model".

Line 346 --> delete it is possible

It would be interesting to introduce the role played by the vascular system in the connection between mammary and gut microbiota. 

The conlusion is long and does not reach any end. They provide a lot of data but do not draw their own conclusions. 

English editing should be revised 

Reviewer 2 Report

In this manuscript, the authors illustrate what is known about the association between the microbiota composition in the gastrointestinal tract and breast cancer progression. Although some questions remain unresolved and need to be addressed, understanding this intricate interplay holds promises for developing innovative therapeutic approaches.

The manuscript is interesting and could give inspiration to subsequent papers.

- The text is clear and well structured.

- The figure clear and legible, the table clear and appropriate.

- The references are updated

I think the manuscript can be published.

- I have a question for the authors: has it been highlighted whether there is a correlation between  gut microbiome and the presence of particular mutations of specific genes? if yes, it would be interesting to add it to the text.

Reviewer 3 Report

There has been enormous interest in the role of the microbiota in cancer progression. This manuscript has shed light on the detailed review of the role of microbiota in breast cancer growth. 

I recommend adding a table on name of microbiota with their role in triple-negative breast cancer.

Please highlight on the strength and weaknesses of the current study.

Round 2

Reviewer 1 Report

The authors has taken into account the suggestions previously made

n/a